# The Surprising Ineffectiveness of Pre-Trained Visual Representations for Model-Based Reinforcement Learning

**Moritz Schneider** [1 2 3 *]   **Robert Krug** [1 2]   **Narunas Vaskevicius** [1 2]
**Luigi Palmieri** [1 2]   **Joschka Boedecker** [3 4]
[1] Bosch Center for Artificial Intelligence   [2] Bosch Corporate Research
[3] University of Freiburg   [4] BrainLinks-BrainTools

## Abstract

Visual Reinforcement Learning (RL) methods often require extensive amounts of data. As opposed to model-free RL, model-based RL (MBRL) offers a potential solution with efficient data utilization through planning. Additionally, RL lacks generalization capabilities for real-world tasks. Prior work has shown that incorporating pre-trained visual representations (PVRs) enhances sample efficiency and generalization. While PVRs have been extensively studied in the context of model-free RL, their potential in MBRL remains largely unexplored. In this paper, we benchmark a set of PVRs on challenging control tasks in a model-based RL setting. We investigate the data efficiency, generalization capabilities, and the impact of different properties of PVRs on the performance of model-based agents. Our results, perhaps surprisingly, reveal that for MBRL current PVRs are not more sample efficient than learning representations from scratch, and that they do not generalize better to out-of-distribution (OOD) settings. To explain this, we analyze the quality of the trained dynamics model. Furthermore, we show that data diversity and network architecture are the most important contributors to OOD generalization performance.

## 1   Introduction

Reinforcement Learning (RL) provides an elegant alternative to classic planning and control schemes, as it allows for complex behaviors to emerge by just specifying a reward, rather than hand-modelling and tuning environments and agents. Despite their success, most methods need extensive data and can be used only on their respective task, lacking the generalization capabilities needed to handle the complexity of real world tasks. On hardware, RL is costly in terms of time and wear, therefore model-based approaches are attractive as they promise to improve sample efficiency. For many real-life problems, vision is an invaluable source of state information, but due to its high-dimensional nature it is challenging to incorporate it in RL algorithms. Therefore, the use of pre-trained visual representations (PVRs) is attractive as, intuitively, it promises to improve sample efficiency and generalization. Most existing approaches use or investigate PVRs in the context of model-free RL. For example, CLIP [1] is already widely used as pre-trained vision model for model-free robotic RL tasks [2, 3, 4]. One would assume that the benefits such representations yield for model-free settings equally apply to model-based methods. In model-based reinforcement learning (MBRL), features of convolutional neural networks (CNNs) are usually used as visual state representations, whereas other representation types such as keypoints, surface normals, depth, segmentation and pre-trained

---

*Correspondence to: `moritz.schneider@de.bosch.com`
Project website: `https://schneimo.com/pvr4mbrl`

representations are often ignored. Moreover, model-based methods are usually trained under an objective mismatch as the training process is required to optimize the accuracy of the dynamics model and the overall performance of the agent at the same time [5]. Naturally, this makes the training procedure different and more difficult than in model-free settings.

In this work, we focus on model-based RL and benchmark a set of representative PVRs on a set of challenging control tasks. To this end we want to answer the following question:

i Is MBRL more sample efficient when using PVRs in contrast to learning a representation from scratch?

Furthermore, PVRs are nowadays increasingly often trained on general datasets (e.g. ImageNet [6], Ego4D [7], etc.) and should be reusable in a wide variety of downstream tasks without further fine-tuning on in-domain data. We would like to empower downstream RL algorithms with corresponding generalization capabilities. Most existing implementations only investigate the distribution shift for the PVRs, but not for the downstream RL algorithm. This leads us to the additional questions:

ii Can model-based agents generalize better to out-of-distribution (OOD) settings with PVRs (i.e. can PVRs pass on their generalization capabilities to model-based agents)?

iii How important are different training properties like data diversity and network architecture of a PVR for model-based agents on a downstream RL task?

Compared with the model-free RL approach, MBRL methods learn accurate models of the environment for efficient learning and planning. Therefore, additionally we want to investigate the final question:

iv How does the quality of the learned dynamics models in terms of accumulation errors and prediction quality differ between models trained from scratch and those based on PVRs?

**Contributions.** The key contributions of this paper are summarized as follows:

- **Benchmarking PVRs for MBRL.** Using PVRs trained on diverse and general data, we study the generalization capabilities to out-of-distribution (OOD) settings of model-based agents utilizing these PVRs. To the best of our knowledge, we perform the first such comparison for MBRL.

- **OOD Evaluation for MBRL and PVRs.** Most other benchmarks only look into the case that PVRs should facilitate better training performance. We additionally look into the case of shifting the distribution also for the underlying MBRL agent, i.e., we have large differences between training and evaluation set. In this way, we investigate to what extent PVRs transfer their generalization capabilities to downstream RL agents. Our experiments reveal that PVRs are often ineffective for the MBRL agents. Furthermore, agents using representations learned from scratch outperform the PVR-based agents most of the time.

- **Important OOD Properties of PVRs.** We investigate and discuss important properties of PVRs for generalization in downstream control tasks in a MBRL context. We find that data diversity and network architecture are the most important contributors to OOD generalization performance.

- **Analysis of Model Quality.** To explain our results more in depth, we analyze the quality of the trained world models and find that those which use representations learned from scratch are in general more accurate and have less accumulation errors regarding predicted rewards than the ones using PVRs.

## 2   Related Work

**Model-Based Reinforcement Learning** (MBRL) combines planning and learning for sequential decision making by using a (learned) predictive model of the environment, a learned value function and/or a policy. These methods have shown impressive results in various domains [8, 9, 10, 11]. However, MBRL is often applied to problems featuring complete state information derived from proprioception [12, 13], which may not always be available in practical scenarios like robotics. Also,

because of the data efficiency of those methods, MBRL on images has been explored extensively [14, 15, 16, 17, 10, 11, 18]. These algorithms typically learn representations from scratch and while there has been research on the influence of reward and image prediction in MBRL [19], we currently do not know any method or review that combines PVRs with MBRL. The only recent trend in a similar direction is action-free pre-training of world models for MBRL [20, 21] and offline MBRL [22, 23, 24]. Both directions pre-train more or all parts of a MBRL algorithm instead of the representation only.

**Visual Representation Learning for Control.** Representation learning is a performance bottleneck in visual reinforcement learning. Pre-training methods and methods using auxiliary tasks offer the potential for greater sample efficiency. As a result, visual representation learning for control especially has received increasing attention recently. In this line of work, RAD [25] enhances the generalization performance and sample efficiency of RL algorithms using visual representations by simply integrating different data augmentations. Concurrently, DrQ [26] and DrQ-v2 [27] integrate similar data augmentations for off-policy RL. Likewise, CURL [28] applies data augmentations to reinforcement learning as well, but uses an additional contrastive auxiliary loss to train the representations. In contrast to the methods by Laskin et al. [25], Yarats et al. [27] and Yarats et al. [26] which learn representations from scratch, we study the use of pre-trained representations on out-of-domain datasets. Pie-G [29] simply uses PVRs pre-trained on ImageNet [6] and shows that using early features of the PVR in combination with BatchNorm [30] results in larger performance gains than using the full PVR. Nair et al. [31] train representations specifically for robotics using a combination of video-language alignment and time-contrastive learning. Value-Implicit Pre-Training (VIP) [32] is a method that combines unsupervised pre-training and reinforcement learning, where a value function is learned implicitly through a self-supervised task, leading to improved performance in downstream reinforcement learning tasks. Radosavovic et al. [33], Xiao et al. [34] and Majumdar et al. [35] pre-train representations based on Masked Autoencoding [36]. While there are combinations which couple representation and reinforcement learning specifically focusing on generalization [25, 28, 26, 27], we focus on general pre-trained representations.

**Benchmarking PVRs in RL and Robotics.** Incorporating PVRs into model-free RL agents has emerged as a pivotal strategy for enhancing the efficiency and generalization capabilities of autonomous agents in visually rich environments. Many studies already investigate different types of pre-trained visual representations for reinforcement learning, robotics or control in general: Sax et al. [37] and Chen et al. [38] benchmark mid-level representations like segmentation and depth estimation models for navigation and manipulation tasks. Wulfmeier et al. [39] analyse dimensionality, disentanglement and observability of a small amount of PVRs and their usefulness as auxiliary task generators in a robot learning setting. Majumdar et al. [35] evaluate a handful of PVRs on a large amount of tasks and introduce a new PVR, VC-1, which we include in our evaluation. Parisi et al. [40] investigate the influence of PVRs on the performance of policies using imitation-learning in a variety of tasks. Similarly, Hu et al. [41] study PVRs using multiple policy learning methods like model-free RL, behavior cloning and imitation learning with visual reward functions. Tomar et al. [42] study the influence of different components for visual RL including world models. They show that MBRL agents using image reconstruction losses fail in diverse environments. In contrast to our approach, they do not study the influence of PVRs on the performance of MBRL. Träuble et al. [43], Chen et al. [38] and Burns et al. [44] are the only works we are aware of that investigate robustness and performance of policies trained using PVRs in OOD settings. The work in Hansen et al. [45] is closest to ours since the authors present similar results in the imitation learning and model-free RL setting by using data augmentation. But all of the papers mentioned previously, use either imitation learning or model-free RL. None of them specifically targets MBRL. In contrast to model-free RL or imitation learning methods, MBRL methods are usually trained under an objective mismatch [5]. Due to this mismatch, we believe in the importance of a focused study on the use of PVRs in MBRL.

## 3 Experiments Setup

### 3.1 Model-Based Reinforcement Learning

As downstream RL algorithms we utilize DreamerV3 [11] and TD-MPC2 [46] which achieve state-of-the-art performance in many benchmarks and are often used in the field [47, 48, 49, 18, 50, 51, 52]. An overview of the algorithms and how we integrated the PVRs is shown in Figure 1.

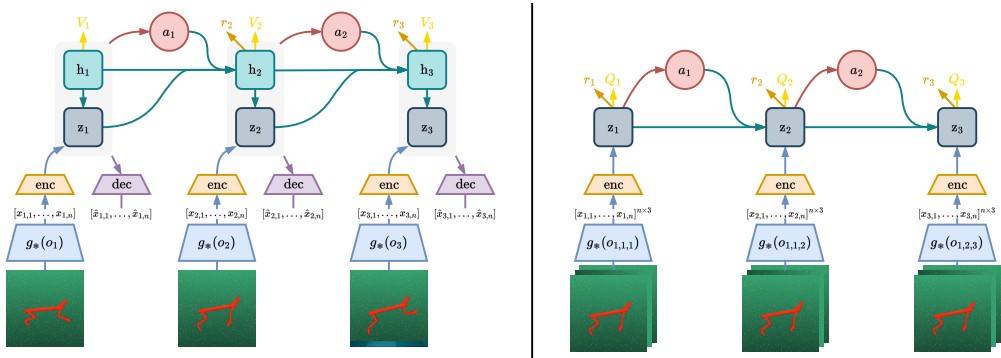

Figure 1: **Components of our PVR-based DreamerV3 (left) and TD-MPC2 (right) architectures.** In DreamerV3, the output $x_t$ of the frozen pre-trained vision module $g_*$ is given to the encoder $\text{enc}(z_t|x_t)$ which maps its input to a discrete latent variable $z_t$. In TD-MPC2 a stack $x_{t-3:t}$ of the last 3 PVR embeddings is given to the encoder $\text{enc}(x_{t-3:t})$ which maps the inputs to fixed-dimensional simplices. The encoder of DreamerV3 additionally requires the recurrent state $h_t$ as input. The rest of both algorithms remains unchanged. Adapted from Hafner et al. [11] and Hansen et al. [46].

In the original setup without PVRs, both algorithms receive a (stack of m) (visual) observation(s) $o_t \in \mathbb{R}^{k \times k \times 3m}$ from the environment, map it to a (discrete) latent variable $z_t \in \mathbb{R}^l$ using an encoder $z_t = \text{enc}(o_t)$. In both cases, $z_t$ can be unrolled into future states $z_{t+1}$ using a latent dynamics model. Based on $z_t$, both methods utilize additional reward and value models for planning and learning. DreamerV3 additionally utilizes a decoder to project latent states back into the observation space. Both MBRL algorithms learn policies in an actor-critic style. TD-MPC2 uses a model-predictive control (MPC) planner in combination with the policy. To study the different PVRs and to retain as much as possible of the original algorithms, we replace the encoder $\text{enc}(z_t|o_t)$ partly with a frozen PVR $g_*(o_t)$. To keep as much information as possible of the PVR encoding $x_t = g_*(o_t) \in \mathbb{R}^n$, we decided to use a linear mapping for the encoder. This decision should reflect current promises of PVRs which state that methods using state-of-the-art representation learning are able to solve downstream tasks using a single linear layer only[1] [53]. The linear mapping is trained jointly with the rest of the MBRL algorithm. Otherwise, the algorithms are kept unchanged and we use mostly the same hyperparameters as the original implementations. For more details we refer to Appendix A.1 and A.2.

## 3.2 Pre-Trained Visual Representations

We refrain from utilizing proprioceptive information (such as end-effector poses and joint positions) to ensure a fair comparison among vision models that solely rely on visual observations. Generally we use the largest published model of each PVR. For more details we refer to Appendix A.3.

We chose to investigate a variety of PVRs, some of which are popular in the field of policy learning (like CLIP) whereas others are less considered in other works (e.g. mid-level representations). Most of them are trained on self-supervised objectives and use either Vision Transformers (ViT) [54] or ResNets [55]. Furthermore, all of them are open-source and easily available. Specifically, we include the following models: CLIP [1], R3M [31], Taskonomy [56], VIP [32], DINOv2 [53], OpenCLIP [57], VC-1 [35], R2D2 [58]. A more in-depth discussion and overview can be found in Appendix A.3.

We additionally include our own pre-trained autoencoders that are trained on task data. This allows us to investigate the influence of the pre-training data on the performance of the PVRs. During pre-training, the autoencoders see the same distribution of data as the other approaches during the reinforcement learning procedure and thus they should have a significant advantage in the subsequent reinforcement learning phase (in which only the encoder is used).

---

[1]In Appendix C we show that the performance differences between multiple nonlinear layers and a single linear layer are negligible. A single linear layer is therefore sufficient.

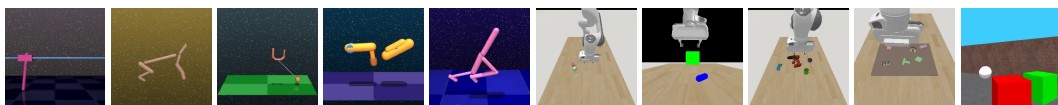

Figure 2: **Illustration of tasks** ranging from DMC and ManiSkill2 to Miniworld with randomizations. Note that while DMC and Miniworld task images show the perspective of the agents, agents in ManiSkill2 tasks utilize the perspective of a wrist-mounted camera.

### 3.3 Domains

We evaluate all representations across a total of 10 diverse control tasks from 3 different domains: DeepMind Control Suite (DMC) [59], ManiSkill2 [60] and Miniworld [61]. All environment observations consist of $256 \times 256$ RGB images, which corresponds to the resolution used for all pre-trained vision models. Most PVRs crop those images down to $224 \times 224$. An overview of the environments and tasks is given in Figure 2.

The agents are trained under a distribution of visual changes in the environment (which we refer to as *In Distribution* (ID) and are evaluated later under a different distribution of unseen changes (OOD changes). ID training and OOD evaluation are implemented through randomizations of visual attributes in the environments by splitting all possible randomizations into ID training and OOD evaluation sets. We focus exclusively on the setting of visual distribution shifts.

We train instances of each agent with different random seeds each performing 12 evaluation rollouts in the training environment every 50000 environment steps during training. For ID and OOD evaluation we perform 200 episode rollouts for each instance after training, resulting in 1200 episodes for each representation per environment. We train DMC agents for 3 million and ManiSkill2 as well as Miniworld agents for 5 million environment steps. Furthermore, TD-MPC2 agents are trained with a smaller set of PVRs on the DMC tasks due to the high computational costs of the experiments and the algorithm.

**DeepMind Control Suite (DMC)** [59] is a widely used and applied benchmark environment for continuous control based on the MuJoCo simulator [62]. It includes locomotion as well as object manipulation tasks in which the agent applies low-level control to the environment. We consider five tasks from the suite: `Cartpole-Swingup`, `Cheetah-Run`, `Cup-Catch`, `Finger-Spin`, and `Walker-Walk`.

To measure ID and OOD performance, we slightly adapt the original tasks from the *DMControl Generalization Benchmark* [63] by changing the background colors of the tasks. Furthermore, we randomize all dimensions of the simulation geometries randomly around their initial values. More specifically, we sample the size values uniformly in a range of $[0.7 \times l_{\text{org}}, 1.3 \times l_{\text{org}}]$ of the original simulation value $l_{\text{org}}$. The OOD evaluation set is a held-out set of 20% of all colors and sizes included in the *DMControl Generalization Benchmark*. The ID training set therefore consists of 80% of the colors and sizes.

Even though the Deepmind Control Suite represents a standard benchmark in RL, none of the PVRs are trained on a visual data distribution similar to DMC providing an even stronger OOD generalization test.

**ManiSkill2** [60] is a suite of robotic manipulation tasks based on the Sapien simulator [64]. The tasks are more challenging than those of DMC, due to their contact-rich nature. Many of the tasks include cluttered and diverse environments and thus are more suitable for testing the generalization capabilities of PVRs. Since many of the PVRs are trained on robotic manipulation data, those tasks represent visually easier shifts than DMC, as a distribution shift still exists but is not as large as in aforementioned DMC tasks. Nevertheless, the tasks are still challenging due to the necessity of precise control skills. We consider four tasks from the suite: `StackCube`, `PlugCharger`, `PickClutterYCB`, and `AssemblingKits`.

Similar to our DMC experiments, we randomize different aspects of the tasks to differentiate between ID and OOD. For `StackCube` we randomize size and color of the cubes but leave the semantic meaning of the colors to solve the task untouched (i.e. picking up a red cube and placing it onto a green one is still the task goal). For `PlugCharger` we randomize shape, size and color of the charger and wall. For both tasks we exclude 20% of the possible randomizations from training and use them for OOD evaluation. `PickClutterYCB` and `AssemblingKits` use diverse combinations

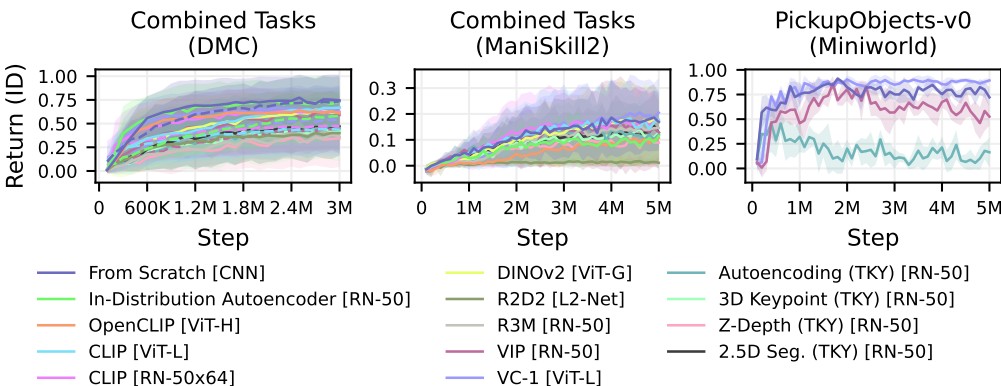

Figure 3: **Normalized ID performance and data-efficiency comparison** on DMC, ManiSkill2 and Miniworld environments between the different representations. Each line represents the mean over all runs with a given representation, the shaded area represents the corresponding standard deviation. Solid lines represent DreamerV3 runs, whereas dashed lines indicate TD-MPC2 experiments. Especially in the DMC experiments, representations trained from scratch outperform all PVRs also in terms of data-efficiency. Curves of each environment individually can be found in Appendix D.

of different objects and object positions and are therefore already diverse by themselves. Thus, we split all possible configurations of positions and objects into a training and evaluation set without additional changes. We train on 80% of the configurations and evaluate OOD on the remaining 20% of configurations. As shown in Hsu et al. [65], it is beneficial to use wrist-mounted cameras instead of third party views. Therefore, we use wrist-mounted camera observations for all ManiSkill2 tasks. For further information, we refer to Appendix A.4.

**Miniworld** [61] is a multi-room 3D world containing different objects. We consider the PickupObjects-v0 task in which the agent is sparsely rewarded for picking up 5 different objects. The location and orientation of both the agent and targets are randomized. In addition, we randomize the color of the target objects. During OOD evaluation the agent is confronted with unseen object colors. We selected the PickupObjects-v0 environment because, unlike the DMC and ManiSkill2 tasks, it utilizes discrete actions and the navigation-based nature implies a dynamically changing background.

## 4 Results

Here we present the results of our evaluation and answer the research questions outlined in Section 1. We start with a general comparison of the data efficiency of PVRs in MBRL (Section 4.1). Afterwards, we evaluate the OOD generalization of PVRs in MBRL (Section 4.2). Furthermore, we investigate which properties of PVRs are important for OOD generalization (Section 4.3). Finally, we analyze the prediction quality of the world models in order to explain our results before (Section 4.4). For better comparisons throughout domains, returns are normalized as recommended by Agarwal et al. [66]. For further information, we refer to Appendix A.5.

### 4.1 Data Efficiency

In the following we want to answer research question i and investigate the sample-efficiency of a PVR-based MBRL agent against the same agent using a visual representation learned from scratch. In general, one would expect that agents using PVRs are more data efficient than their counterparts with representations trained via reinforcement learning. The common assumption is that PVRs are able to capture relevant information of the environment due to their pre-training phase. Therefore, the downstream learning algorithm can focus on learning the dynamics of the environment. This ought to be especially pronounced for visual foundation models, which promise to generalize to different domains [67]. The results for our MBRL setting are summarized in Figure 3.

Surprisingly, representations learned from scratch are in most cases equally or even more data-efficient than the PVRs. We want to highlight that this is also true for autoencoders which are pre-trained

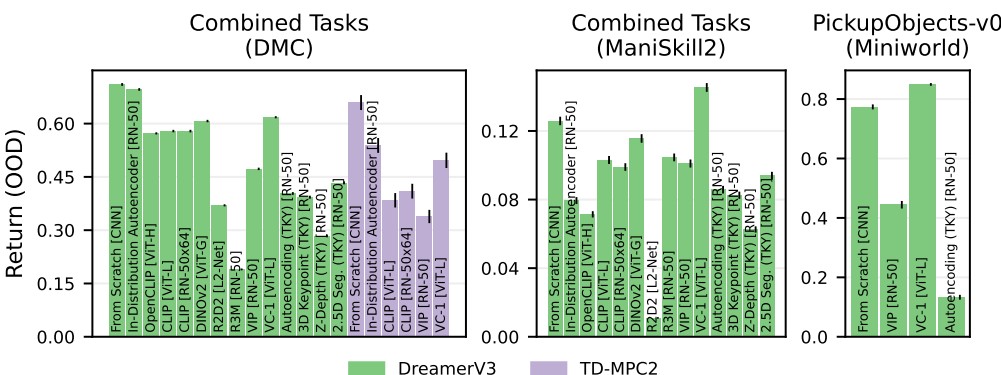

Figure 4: **Average normalized performance on DMC, ManiSkill2 and Miniworld tasks in the OOD setting.** The baseline representation learned from scratch outperforms all PVRs, even in the OOD settings. Thin black lines denote the standard error.

in-distribution on our own task-specific data. This contradicts the general belief that PVRs accelerate the training of (MB)RL agents [38]. We hypothesize that this is due to the objective mismatch in MBRL [5]. The overall optimization is decoupled into two optimization procedures (one for the dynamics and one for the policy/reward/value), making it harder to adapt to an existing representation instead of learning a new one from scratch.

## 4.2 Generalization to OOD Settings

Even if PVRs are not able to perform equally well as representations learned from scratch in the ID case, they might be able to perform better in OOD domains. This should be especially true for visual foundation models which are often trained on diverse data [1, 68, 53, 69, 57]. Therefore, here we want to answer research question ii and evaluate the OOD performance on both domains after training. The results are visualized in Figure 4. With the exception of VC-1, it is noticeable that no PVR performs good in both domains. Even autoencoders trained in-distribution on task-specific data perform worse compared to training an encoder from scratch. This is surprising, since some PVRs are trained on large sets of diverse data and should therefore generalize well to OOD domains which, however, we find not to hold true when compared to agents with representations learned from scratch. This is especially evident for the DMC environments where the PVRs perform worse than training from scratch.

## 4.3 Properties of PVRs for Generalization

The results so far show that PVRs perform not as well in MBRL and for OOD generalization as they do in policy learning settings [40, 35, 41]. Also, the results do not explain which properties of the different PVRs are relevant for OOD generalization. This is the subject of our next research question iii. Based on Table 2, we group the PVRs into categories and evaluate the ID and OOD performance of the PVRs in each category. The exact categorization can be found in Appendix B. The results are depicted in Figure 5 and discussed below.

**Language Conditioning.** All CLIP-based PVRs as well as R3M are conditioned on language in their training procedure. However, the combined performance of these PVRs shown in Figure 5 indicates that language conditioning is not necessary for good OOD generalization. This is surprising, since language conditioning is a popular technique to improve capabilities of vision-based agents [2, 70, 31, 4, 71, 72, 48] and is assumed to be a strong bias for a visual model as it should provide semantically relevant features [31]. Nevertheless, our results suggest that pre-training representations with language might not be as helpful for control tasks as it might be for other direct downstream learning tasks.

**Sequential Data.** Reinforcement learning is a sequential decision making problem. Thus, a good representation should capture the sequential dynamics of the environment. This is even more relevant for MBRL, where the representation is additionally used to predict the state evolution. Therefore,

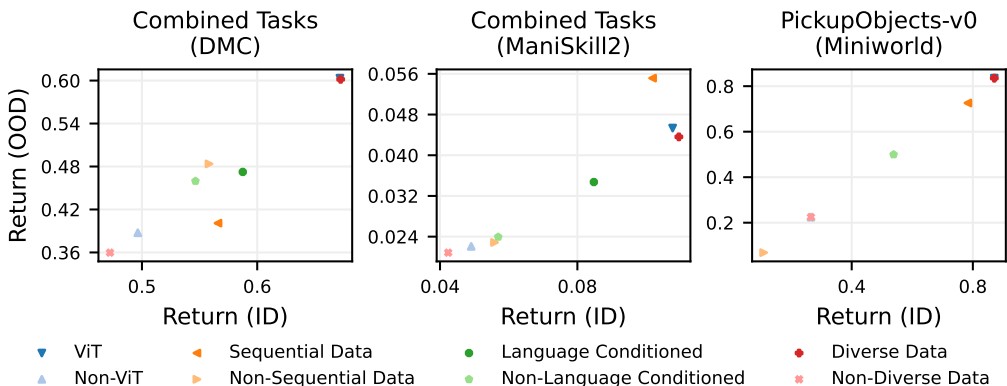

Figure 5: **IQM return of the different categorizations.** Each marker represents the interquartile-mean performance of an individual group. The x-axis shows the ID performance and the y-axis the OOD performance. Especially, ViT representations or representations trained on diverse data perform well in the OOD setting. Sequential data seem to help in ManiSkill2 and Miniworld but not in DMC. Categorization plots for each environment individually can be found in Appendix E.

this category includes PVRs which are trained on sequential data. The results in Figure 5 show that the sequential order of the data is somewhat beneficial (especially in the ManiSkill2 experiments).

**Data Diversity.** Foundation models are well known to be trained on diverse data [70, 71, 68, 1, 72]. We define data diversity based on the number of datasets used to train a PVR and/or based on the size of a dataset (e.g. WIT is supposed to be diverse). If a PVR is trained on more than one dataset or data domain we assume the data to be diverse. The results indicate that data diversity is generally important for performance in both task domains.

**Vision Transformer Architecture.** In recent years, ViTs have gained attention as a powerful alternative to ResNets. In contrast to the other networks, ViTs show good performance on both domains which aligns with results from Burns et al. [44]. However, we find that the ViT architecture is not the only relevant factor for good OOD generalization. For example, CLIP [ViT-L] and OpenCLIP are both based on the ViT architecture, but both perform on par with CLIP [RN50x64]. This is suprising, since the ViT-based CLIPs are trained with another structure and the same loss, but with different data. This indicates that data diversity might be more important than the network architecture.

### 4.4 World Model Differences

According to research question iv, we investigate how the different visual representations influence the quality of the world model which is learned in the downstream MBRL algorithm. To this end, we train model-based agents on the `Pendulum Swingup` task of DMC with the same setup as described in Section 3.3. We then use a pre-collected dataset of 200 diverse trajectories to analyze the world models of the agents. We plot the error of the complete trajectory and a smaller window of the same trajectories with a horizon of 33 timesteps to show the differences between long-term and shorter predictions.

**Dynamics Prediction Error.** For planning purposes, MBRL algorithms employ a forward dynamics model of the environment. The model can either be given [8, 9] or learned [16, 73, 11, 74, 46]. DreamerV3 as well as TD-MPC2 belong to the latter category and we can therefore analyze the dynamics prediction error of the underlying models. The results are shown in Figure 6. The plots show that the state evolution prediction accuracy of PVR-based approaches is comparable to model-based agents using representations learned from scratch. Furthermore, we observe a slight negative correlation between the values presented in Figures 4 and 6 ($r = -0.22$, $p = 0.4$). This suggests that the quality in the dynamics prediction plays a minor role in the performance since the models with the best task performance do not necessarily have the lowest dynamics prediction error. Thus the dynamics prediction error is not the only important factor for the performance of the agent.

**Reward Prediction Error.** The reward model is a crucial part of the world model. Because state-of-the-art model-based algorithms are actor-critic based (as in our case), they use learned value

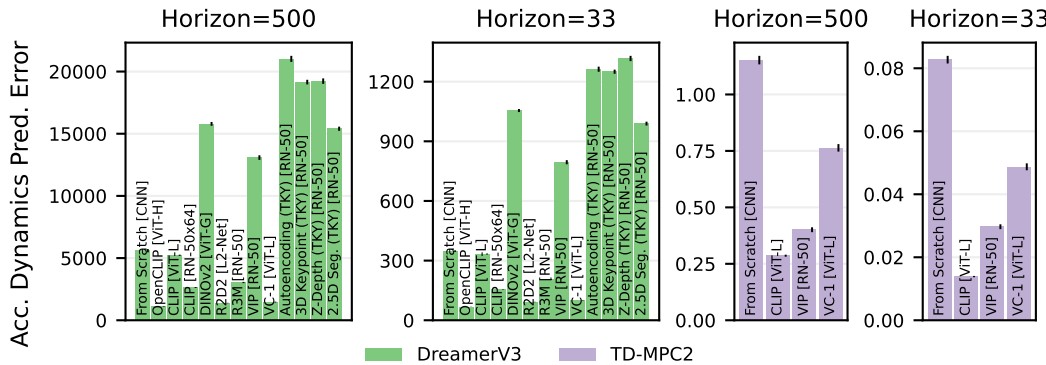

Figure 6: **Average Accumulated Dynamics Prediction Errors** on the `Pendulum Swingup` task for 200 trajectories. For DreamerV3 we average the forward and backward KL divergence between the prior and posterior distributions of the latent state $z_t$. For TD-MPC2 the MSE between the predicted latent state $\hat{z}_t$ of the dynamics model and the encoded latent state $z_t$ is plotted. Thin black lines denote the standard error.

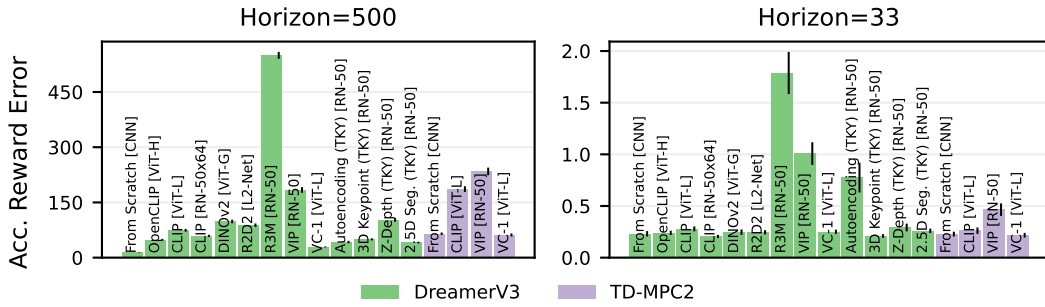

Figure 7: **Average Accumulated Reward Errors** on the `Pendulum Swingup` task for 200 trajectories. The error is calculated as the absolute difference between true and predicted reward $|r_t - \hat{r}_t|$. Thin black lines denote the standard error.

functions to update the policy. Since the value prediction depends upon the the reward prediction, the agent might not be able to learn a good policy if the reward prediction is inaccurate. Figure 7 shows an analysis of the models reward prediction errors. The plots demonstrate that models which have a better task performance (e.g. from scratch or VC-1) generally produce more accurate reward predictions. Here we observe a significantly stronger negative correlation between task performance and reward error compared to the dynamics prediction case ($r = -0.66$, $p = 0.004$). This indicates that the reward prediction is generally important for the performance of the agent, and that methods which perform badly on the tasks are unable to predict future rewards accurately. The PVRs do not appear to provide enough information to predict the reward better than agents without PVRs do. But as can be seen in the specific case of 2.5D Segmentation (TKY) it becomes clear that low reward prediction errors do not imply good performance necessarily. We assume that the influence of other components is still important.

**Reward Information in the Latent State Space.** PVRs are trained to compress information since the representations are trained as information bottlenecks. As such the training might not capture reward-relevant data from the pre-training images since reward information is usually not a component in the training objective of the PVRs. On the other hand, MBRL methods like DreamerV3 and TD-MPC2 heavily rely on reward information in their objectives. Furthermore, previous benchmarks on PVRs often examine imitation learning settings where reward information is usually irrelevant. Based on our previous results we therefore examine the extractable reward information content in the representations. In Figure 8 we show UMAP [75] projections of the latent state space of model-based agents for different representations in the investigated `Pendulum Swingup` task. For representations learned from scratch, the reward information in the latent state space projection is highly structured and states with similar rewards are more closely embedded in the latent space of representations that perform better in DMC. We hypothesize that this is a contributing factor to why many PVRs

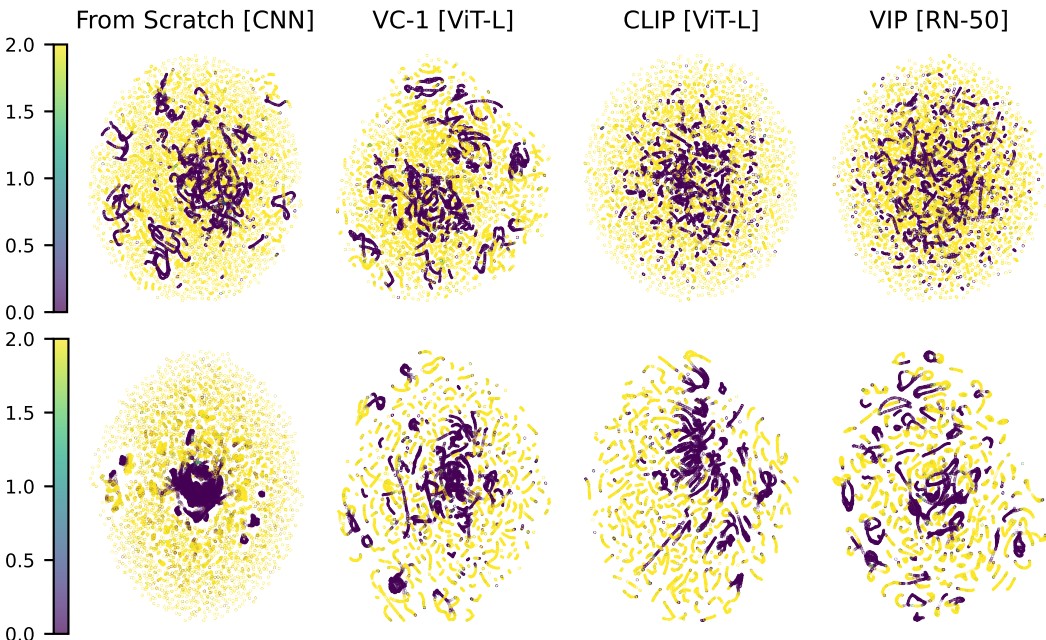

Figure 8: **UMAP projections of DreamerV3 (top row) and TD-MPC2 (bottom row) encodings** using different representations as input. The points are color coded by the real perceived reward. Each point represents a visited state in the `Pendulum Swingup` environment of DMC. The representations learned from scratch better disentangle low and high reward states whereas the embeddings of the PVRs are more entangled.

underperform compared to representations learned from scratch. In order to learn a good policy, the agent needs to be able to extract and predict reward information accurately. This is not possible if the reward information is not consistently embedded in the representation.

## 5 Conclusion

In this work, we evaluate the efficiency and generalization capabilities of different PVRs in the context of model-based RL. We provide empirical evidence that PVRs neither improve sample efficiency of model-based RL, nor ultimately empower MBRL agents to generalize better to completely unseen out-of-distribution shifts. Experiments analyzing the quality of the trained world model suggest that model-based agents utilizing PVRs are not able to learn good reward models for the tasks compared to agents learning representations from scratch. This indicates that PVR-based approaches can learn comparable dynamics models but struggle to learn good reward models which are crucial for the performance of MBRL agents. We conclude that PVRs might lack the needed information for reward model learning, and that training visual representations for MBRL requires extra attention compared to model-free RL. Additionally, we conducted experiments to find relevant performance properties of PVRs. Here, diversity in the pre-training data as well as a ViT architecture seem to improve generalization capabilities to OOD settings of PVRs. Conversely, a language-conditioned loss or sequential training data seem to play minor roles.

**Limitations.** This paper aims not to be the final word on the topic of OOD generalization of PVRs, RL and MBRL in particular. Our findings hopefully inspire more researchers to dig into the untapped potential of utilizing PVRs in MBRL, since more experiments are needed to draw a final conclusion. Naturally, a benchmark like the one presented in this work is inevitably computationally demanding. It was therefore necessary to make certain design decisions and restrict the number of representations. From our point of view, we focused on the most important PVRs but other MBRL algorithms and PVRs are certainly of interest as well. Furthermore, we evaluate the PVRs on 3 domains. Experiments on other domains, especially in the real-world are needed.

## Acknowledgments and Disclosure of Funding

Joschka Boedecker is part of BrainLinks-BrainTools which is funded by the Federal Ministry of Economics, Science and Arts of Baden-Württemberg within the sustainability program for projects of the excellence initiative II.

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

# A Implementation Details

We run all our experiments on a compute cluster using 6 to 8 cores and 32GB memory per experiment with either a single NVIDIA V100 or A100 GPU depending on the PVR used. Depending on the representation/input size more memory might be needed (e.g. 96GB for the ManiSkill2 from scratch experiments).

## A.1 DreamerV3

For each task and encoding type (i.e. PVR or from scratch) combination we train 6 instances of DreamerV3 for DMC and ManiSkill2 with another random seed each resulting in 756 trained instances. For Miniworld, we train 4 instances of DreamerV3 with another random seed each resulting in 16 additional trained instances. The PVR networks are implemented as environment wrappers taking image observations $o_t$ as input while returning the representation $x_t$. The representation is then fed into DreamerV3 and is stored for further training in the replay buffer. Thus, the original DreamerV3 algorithm works solely on the representations $x$ as input using an additional MLP or linear layer. As a result, the algorithm decodes the encoding $x$ only and not the whole input image $o$. To keep the encoder-decoder structure of DreamerV3, we replace the original encoder partly only instead of removing it fully. We use the implementation from https://github.com/danijar/dreamerv3 (MIT license). Hyperparameters are shown in Table 1 but in most cases do not deviate from the implementation.

Table 1: Overview of the hyperparameters for DreamerV3.

|  | DMC | ManiSkill2 | Miniworld |
|---|---|---|---|
| Corresponds to model size in Hafner et al. [11] | S | XL | XL |
| **General** | | | |
| Replay Capacity | $10^6$ | $10^6$ | $10^6$ |
| Batch Size | 16 | 16 | 16 |
| Batch Length | 64 | 64 | 64 |
| Start Learning (Prefill) | 0 | 0 | 0 |
| Action Repeat | 2 | 1 | 1 |
| Environment Steps | $3 \times 10^6$ | $5 \times 10^6$ | $5 \times 10^6$ |
| Train Ratio | 512 | 32 | 32 |
| **Actor Critic** | | | |
| Imagination Horizon | 15 | 15 | 15 |
| Discount | 0.95 | 0.95 | 0.95 |
| Return Lambda | 0.95 | 0.95 | 0.95 |
| Learning Rate | $3 \times 10^{-5}$ | $3 \times 10^{-5}$ | $3 \times 10^{-5}$ |
| Adam epsilon | $1 \times 10^{-5}$ | $1 \times 10^{-5}$ | $1 \times 10^{-5}$ |
| Gradient Clipping | 100 | 100 | 100 |
| **World Model** | | | |
| RSSM Size | 512 | 4096 | 4096 |
| Number of Latents | 32 | 32 | 32 |
| Classes per Latent | 32 | 32 | 32 |
| Learning Rate | $1 \times 10^{-4}$ | $1 \times 10^{-4}$ | $1 \times 10^{-4}$ |
| Adam epsilon | $1 \times 10^{-8}$ | $1 \times 10^{-8}$ | $1 \times 10^{-8}$ |
| Gradient Clipping | 1000 | 1000 | 1000 |
| **Plan2Explore** | | | |
| Ensemble Size | 10 | 10 | 10 |
| $\beta$ | 1 | 1 | 1 |

**Symlog.** The DreamerV3 baseline instances symlog their inputs as described in the original paper. For the ManiSkill environments we found that not applying symlog to the PVR-based instances

performs better. For the other environments we keep applying symlog for both baseline instances and PVR-based instances.

**Exploration Reward.** We use an additional exploration reward $r_t^{\text{expl}}$ in all experiments which changes the per-timestep reward $r_t$ for the agent to

$$r_t = r_t^{\text{env}} + \beta r_t^{\text{expl}} \tag{1}$$

where $r_t^{\text{env}}$ is the original reward from the environment and $\beta$ determines the amount of exploration. We use Plan2Explore [76] to calculate $r_t^{\text{expl}}$ with an ensemble size of 10 and $\beta = 1$. This helps in the overall performance in the environments.

**Encoder Trained From Scratch.** The representations which are trained from scratch are using the unmodified code from the implementation. Due to high computational demands we train the DMC as well as the Miniworld baselines on $64 \times 64$ and the ManiSkill2 baseline on $128 \times 128$ pixels.

## A.2 TD-MPC2

For each task and encoding type (i.e. PVR or from scratch) combination we train 4 instances of TD-MPC2 with different random seeds. We use the implementation from `https://github.com/nicklashansen/tdmpc2` (MIT license). The encoder trained from scratch uses the unchanged implementation.

The PVR encoding is implemented as an environment wrapper which takes image observations $o_t$ as input and returns the representation $x_t$. Similar to the original training from scratch, we stack the last 3 embeddings $x_{t-3:t}$ of a PVR into a single feature vector. The stack is then fed into TD-MPC2 and is stored for further training in the replay buffer. The original TD-MPC2 algorithm works solely on the representations $x$ as input using an additional MLP or linear layer. To keep the encoder structure of TD-MPC2 in the PVR-based approaches, we replace the original encoder partly only instead of removing it fully. The residual hyperparameters do not deviate from the original implementation for visual RL.

## A.3 PVRs

For all representations we use the same preprocessing steps as described by the authors. We use the implementations and the pre-trained models from:

| # | Name | URL | License |
|---|------|-----|---------|
| 1 | CLIP | `https://github.com/openai/CLIP` | MIT |
| 2 | DINOv2 | `https://github.com/facebookresearch/dinov2` | Apache 2.0 |
| 3 | OpenCLIP | `https://github.com/facebookresearch/ImageBind` | CC BY-NC-SA 4.0 DEED |
| 4 | R2D2 | `https://github.com/naver/r2d2` | CC BY-NC-SA 3.0 DEED |
| 5 | R3M | `https://github.com/facebookresearch/r3m` | MIT |
| 6 | Taskonomy | `https://github.com/alexsax/visual-prior` | MIT |
| 7 | VC-1 | `https://github.com/facebookresearch/eai-vc` | CC BY-NC 4.0 DEED |
| 8 | VIP | `https://github.com/facebookresearch/vip` | CC BY-NC 4.0 DEED |

An overview of the PVRs is given in Table 2.

**CLIP** (Contrastive Language-Image Pre-training) [1] is a method to learn a joint representation space for images and text through a contrastive learning approach utilizing 400 million image-text pairs. One CLIP instance leverages two neural networks. A vision encoder network for image understanding and an additional language model for textual comprehension. The method uses a contrastive objective, which encourages both networks to bring similar image-text pairs closer and push dissimilar pairs apart in their shared embedding space. For our experiments we utilize the vision encoder only. More specifically, we use the ResNet-50x64 and the ViT-L models.

**R3M** (Reusable Representations for Robotic Manipulation) [31] tries to learn generalizable visual representations for robotics from videos of humans and natural language. The pre-training of the

Table 2: **Overview of used PVRs** sorted alphabetically and according to the used backbone. In each row we describe the loss, dataset, model architecture and embedding size of a specific PVR (M corresponds to 1 million).

| # | Model | Loss | Dataset | | Backbone | Emb. |
|---|-------|------|---------|---|----------|------|
| 1 | Autoencoder | Mean squared error | In-domain | In-domain data of the respective task | ResNet-50 | 1536 |
| 1 | CLIP | Contrastive image-language pre-training objective | WIT | 400M image-text pairs from the internet | ResNet-50x64 | 1024 |
| 2 | R3M | Time-Contrastive video-language alignment pre-training objective | Ego4D | 5M images from a subset of Ego4D | ResNet-50 | 2048 |
| 3 | Taskonomy | Supervised task-dependent encoder-decoder objective | Taskonomy | 4M images of indoor scenes from about 600 buildings with annotations for every task | ResNet-50 | 2048 |
| 4 | VIP | Goal-conditioned value function pre-training objective | Ego4D | 5M images from a subset of Ego4D | ResNet-50 | 1024 |
| 5 | CLIP | Contrastive image-language pre-training objective | WIT | 400M image-text pairs from the internet | ViT-L | 768 |
| 6 | DINOv2 | Discriminative self-supervised pre-training objective | LVD-142M | 142M images collected from various datasets for classification, segmentation, depth estimation and retrieval | ViT-G | 1536 |
| 7 | OpenCLIP | Contrastive image-language pre-training | LAION-2B | 2000M image-text pairs | ViT-H | 1024 |
| 8 | VC1 | Masked Autoencoding (MAE) | Ego4D-MNI | 5.6M images from Ego4D, manipulation-centric data, indoor navigation data, and ImageNet | ViT-L | 1024 |
| 9 | R2D2 | Unsupervised objective estimating reliability and repeatability of keypoints at the same time | (i) Oxford and Paris retrieval, (ii) Aachen Day-Night | (i) 1M distractor images from Oxford and Paris, (ii) 14067 images with changing conditions (day-night, season, weather) | FC L2-Net | 1024 |

representation utilizes the Ego4D dataset [7] using a combination of time-contrastive learning and video-language alignment. An additional L1 penalty encourages a sparse and compact embedding.

**Taskonomy** [56] is an approach to study and understand the relationships between different visual tasks, e.g. object recognition, depth estimation, keypoint detection, semantic segmentation, etc., with the goal of leveraging shared representations across tasks. The study trains different mid-level visual representations on a variety of visual tasks using a dedicated pre-collected dataset. The works in Sax et al. [37] and Chen et al. [38] show that those mid-level representations can be used in a training pipeline for downstream visual navigation, as well manipulation policies. We utilize a subset of these representations for our experiments, namely *Autoencoding*, *3D Keypoint Detection*, *2.5D Segmentation* and *Z-Depth Estimation*. We denote those representations with an additional *TKY* lettering in the experiments.

The Taskonomy representations we utilize for our experiments use an encoder-decoder-style training objective. The encoder is trained to embed the input image while the decoder is trained to reconstruct the input image into the pixel-task space (e.g. the segmented image).

**VIP** (Value-Implicit Pre-Training) [32] casts representation learning from egocentric human videos as an offline goal-conditioned reinforcement learning problem. The authors formulate a self-supervised goal-conditioned value function objective that does not depend on actions. This enables pre-training on unlabeled egocentric human videos. VIP can be understood as a value-implicit time

contrastive objective that generates a temporally smooth embedding, enabling the value of a state to be implicitly defined via the embedding distance.

**DINOv2** [53] is a method to learn robust visual features from images using a combination of different techniques to scale pre-training in terms of data and model size. Furthermore, DINOv2 utilizes a dedicated, diverse, and curated image dataset of 142 million images. The authors trained a ViT-G model with 1 billion parameters that surpasses the best available all-purpose vision model, OpenCLIP, on most of the benchmarks at image and pixel levels. For our experiments we use the initial ViT-G model.

**OpenCLIP** [57] is an open-source implementation of CLIP [1]. It uses the same paradigm as CLIP, where separate models are used to encode text and images into a single embedding. OpenCLIP models have been trained on a variety of data sources and compute budgets, ranging from small-scale experiments to larger runs. We utilize an OpenCLIP ViT-H model trained on LAION-2B [77].

We utilize an OpenCLIP ViT-H model trained on LAION-2B [77] via the open-source implementation of ImageBind [69]. ImageBind initializes and freezes the image and text encoders using the OpenCLIP model resulting in the same encoders for these modalities.

**VC-1** (VisualCortex-1) [35] aims to support a diverse range of sensorimotor skills and environments by training them on 5.6 million images of the Ego4D-MNI dataset consisting of the original Ego4D data [7] and additional data from manipulation tasks, navigation tasks and the ImageNet dataset [6]. The Masked Autoencoding (MAE) objective from He et al. [36] is utilized. Due to the similarities to Radosavovic et al. [33] and Xiao et al. [34], we do not include those PVRs based on MAE in our evaluation.

**R2D2** [58]. In order to evaluate descriptor representations, we use a pre-trained R2D2 model. This self-supervised method proposes to learn keypoint detection and description in combination with a predictor of the local descriptor discriminativeness. The model predicts a set of sparse locations of an image that should be repeatably and reliably detected. R2D2 is the only PVR that neither uses a ViT nor a ResNet architecture, but instead employs a fully-convolutional L2-Net (FC L2-Net) [78]. The model is trained on Oxford-Paris retrieval [79] and the Aachen-Day Night [80] datasets.

For each processed image we choose the top-8 descriptor vectors regarding reliability, repeatability and spatial distance similar to the *Spatial Descriptor Set* of Manuelli et al. [81]. If an image provides less than 8 descriptors we repeat the found descriptors until 8 descriptors are reached. If no descriptor is found we use a zero-vector as descriptor. Furthermore, instead of shifting multiple times over the image with different sizes we only shift once and calculate the descriptors on the original size.

### A.4 Environments Details

**ManiSkill2.** For the ManiSkill2 environments we integrate an end-effector velocity controller which prohibits rotations around the x- and y-axis and stabilizes these rotations automatically. This results in a 4-dimensional action space ($\text{pos}_x$, $\text{pos}_y$, $\text{pos}_z$, $\text{rot}_z$). To further increase the realism we include a textured table and a textured floor (freely available under CC0 license from `https://polyhaven.com`).

### A.5 Return Normalization

For all our evaluations we normalize returns $G_0$ by

$$\frac{G_{0,agent} - G_{0,random}}{G_{0,maximum} - G_{0,random}} \tag{2}$$

where $G_{0,random}$ is the return of a random policy and $G_{0,maximum}$ is the theoretical maximum achievable return (i.e. the maximum achievable timesteps of the environment for ManiSkill2 and DMC; 200 for ManiSkill2, 1000 for DMC and 5 for Miniworld). The return of the random policy is calculated by averaging the returns of 2500 episodes of the random policy.

# B  Properties Groups

Table 3: Categorization of PVRs according to their properties.

| # | Property | Pre-trained Visual Representations (PVRs) |
|---|----------|-------------------------------------------|
| 1 | ViT | CLIP [ViT], DINOv2, VC-1, OpenCLIP |
| 2 | Sequential Data | R3M, VIP, VC-1 |
| 3 | Language Conditioned | R3M, CLIP [ViT], CLIP [RN-50x64], OpenCLIP |
| 4 | Diverse Data | OpenCLIP, CLIP [ViT], CLIP [RN-50x64], DINOv2, VC-1 |

# C  Linear Layers Versus Multilayer Perceptrons

To ensure that the training does not depend on the choice of encoder size (linear layer vs. MLP), we trained such DreamerV3 instances on a handful environments with VC-1 as well as representations from scratch. Comparing the results in Figure 9 shows that the performance is not dependent on the choice of encoder size and that differences between the MLPs and linear layers are negligible.

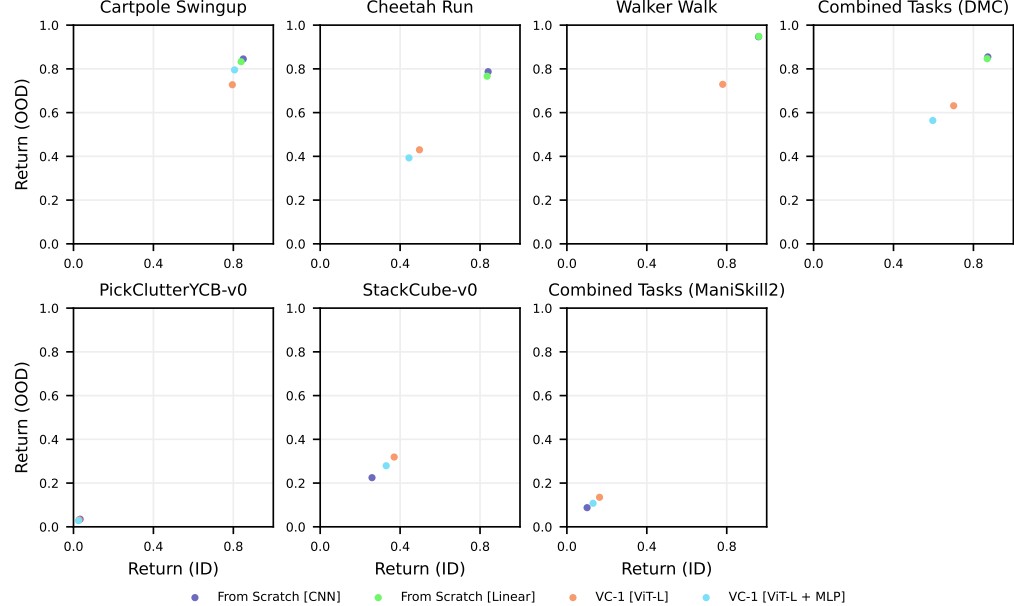

Figure 9: **Performance comparison with representations using linear layers versus multilayer perceptrons.** Top row shows normalized ID and OOD performance on DMC environments whereas the bottom row shows the performance on ManiSkill2 environments. Differences between MLPs and linear layers are negligible.

# D    Performance Curves (All Tasks)

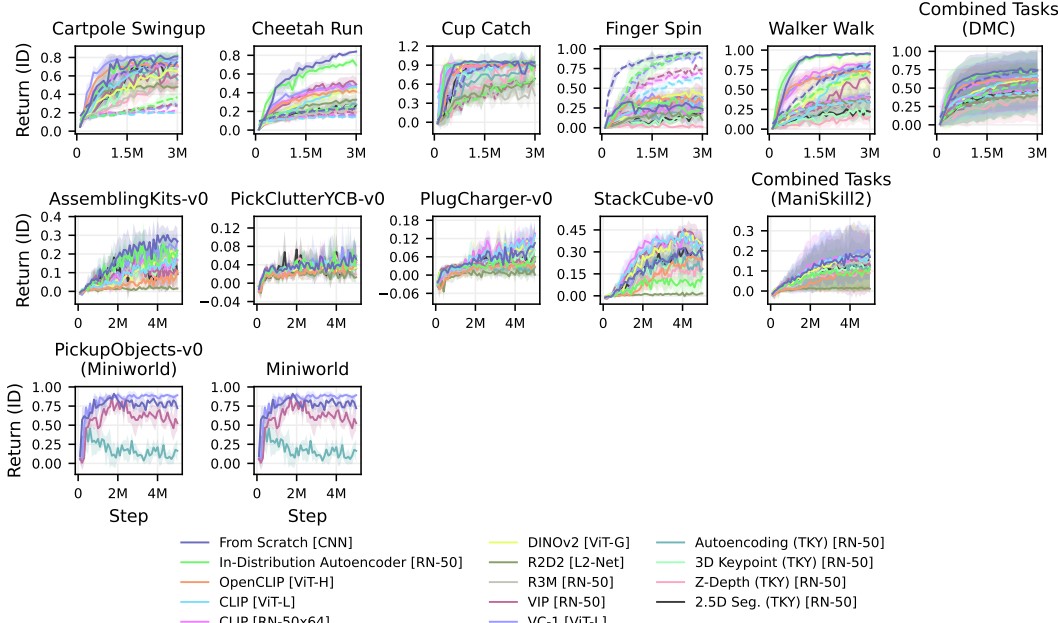

Figure 10: **Performance and data-efficiency comparison** for each task of ManiSkill2, DMC and Miniworld between the different representations. The solid/dashed line shows the mean over multiple runs for DreamerV3/TD-MPC2. The shaded area represents the standard deviation of the respective representation.

# E    Aggregated Performance of Different Properties (All Tasks)

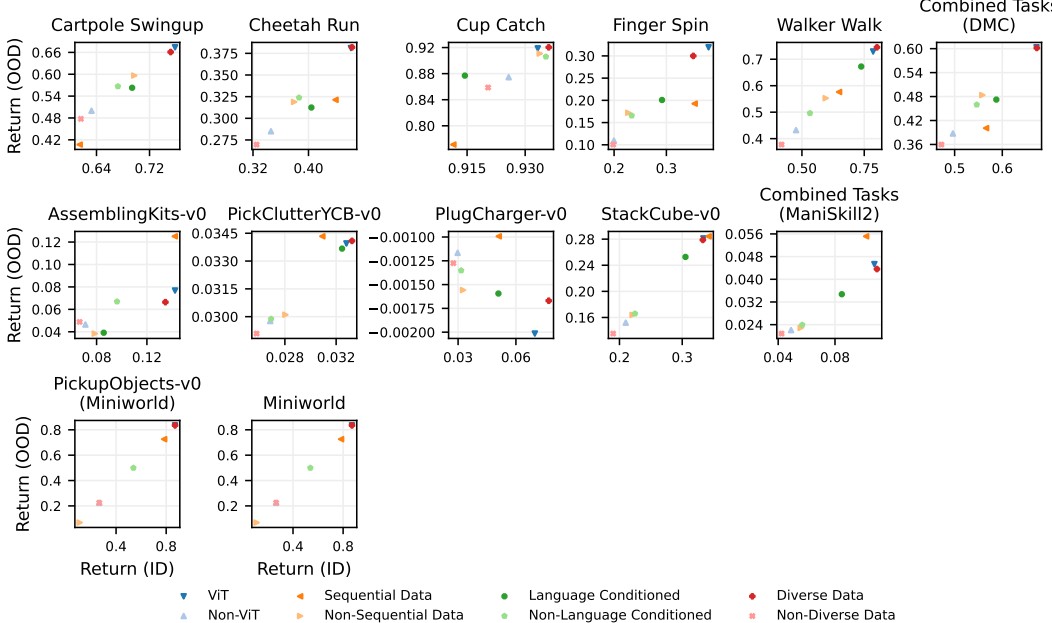

Figure 11: **IQM return of the different properties on task level.** Each marker represents the interquartile-mean performance of an individual group.

# F   Transformer Block Ablations with VC1

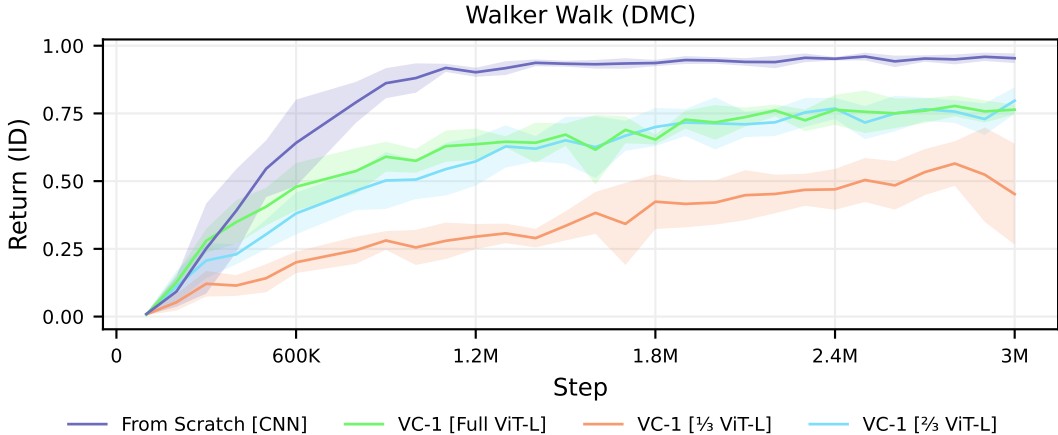

Figure 12: **Ablation of transformer blocks in VC-1.** Using ⅔ of VC-1 results in similar performance compared to the full model. Transformer blocks near the final one seam to offer as much information as the final output. With only ⅓ of VC-1 the performance drops significantly. It seems that earlier representations do not offer enough information for the MBRL agent to perform better or similarly.

