# OpenReview forum: "The Surprising Ineffectiveness of Pre-Trained Visual Representations for Model-Based Reinforcement Learning"
_NeurIPS.cc/2024/Conference — NeurIPS 2024 poster_

### Official Review · Reviewer_32eA · 2024-07-08

**Soundness:** 3
**Presentation:** 3
**Contribution:** 2
**Rating:** 5
**Confidence:** 4

**Summary:**

This paper presents a study on pre-trained visual representations (PVRs) to improve sample efficiency of model-based reinforcement learning methods (MBRL). It studies a range of PVRs with different architectures, pre-training objectives and data modalities on two recent model-based algorithms: DreamerV3 and TD-MPC2. The authors propose to replace image observations with frozen PVRs features, replacing the original CNN encoder/decoder with a trainable linear layer. The paper focuses on 9 diverse continuous control tasks from 2 different domains: The DeepMind Control Suite (DMC) and ManiSkill2. The results show that the use of PVRs surprisingly does not help to improve performance and has diminishing returns compared to training encoder representations from scratch.

**Strengths:**

The paper proposes a very interesting study on the application of PVRs for two recent model-based approaches. The use of PVRs for model-based RL is a promising area of research. The performance and learning speed of model-based agents like Dreamer is very linked to the quality of representations learned by the world model. Benefiting from already pre-trained representations to reduce training time or increase final performance would be very appreciated since the training time of model-based agents can take multiple days for some advanced tasks.

The paper evaluates a diverse range of pre-trained models, showing the normalized returns of each pre-trained model. The authors find that learning the encoder representation from scratch leads to better performance for DreamerV3 and TD-MPC2, contradicting expectations following previous works.

The paper also experiments with out-of-distribution (OOD) versions of the environments (colors, object sizes), finding that PVRs do not help either to improve performance in this setting.

The paper is clear and well written.

**Weaknesses:**

The study on the potential effect of using PVRs on MBRL methods is limited. The paper only studies the use of frozen PVRs, where the world model gradients are not back-propagated to the pre-trained weights during training. Solving RL tasks often requires the agent to pay attention to small details and learn very detailed representations, especially for DMC tasks. It would have been interesting to study the use of PVRs without frozen weights during training.

As mentioned in the conclusion limitations, the authors only study the use of PVRs for continuous control tasks. It would have been interesting to study the application of PVRs to another domain based on discrete actions (Atari games, DeepMind Lab 3D environments, Minecraft...) or more aligned with data seen during PVR pre-training.

**Questions:**

Did the authors experiment with fine-tuning the PVRs during training instead of using frozen representations ? It would have been interesting to experiment with optimizable PVRs (without reconstruction loss in the case of DreamerV3). Maybe the use of PVRs could exempt DreamerV3 from using a reconstruction loss for learning encoder representations. The use of frozen representations may also be the cause of the observed results, all information details required to achieve good performance may not be recoverable from the compressed pre-trained representations.

Did the authors experiment with reconstructing the image observations from frozen PVRs in order to study the possibility of recovering image information from representations ? Also to assess the quality of trajectories imagined by the world model.

Did the authors experiment with using representations of other layers than the final layer? The usefulness of representations of pre-trained models can vary greatly depending on the chosen layer. Are the results different when using the features of inner layers or a collection of layers features ?

For the in-distribution autoencoder PVR, what policy is used to generate the data used for pre-training ? Is it a random policy, a collection of policies to ensure diverse behaviors, or a pre-trained policy ? I ask the question because during training, the PVR should provide representations for the widest distribution of agent trajectories or positions. DreamerV3 has sometimes troubles to reconstruct agent situations that are seen often when training on DMC tasks.

I would be ready to increase my score if additional experiments and analysis are provided by the authors.

**Limitations:**

The authors adequately addressed the limitations in the conclusion.

---

> ### Author Rebuttal · Authors · 2024-08-06
>
> __Questions:__
>
> > “Did the authors experiment with fine-tuning the PVRs during training instead of using frozen representations ?”
>
> We actively decided to omit fine-tuning experiments since the original promise of many of the used representations is to perform (near) SOTA on downstream tasks without additional fine-tuning. Fine-tuning itself might not be an easy task since it can destroy parts of the representation needed for good generalization performance [1]. Ultimately, we believe that (if done right) fine-tuning can boost performance for each of the representations. But we are unsure if the performance boost would outperform training from scratch. This aligns with mixed results from end-to-end fine-tuning from the original VC-1 paper [2].
>
> > "Maybe the use of PVRs could exempt DreamerV3 from using a reconstruction loss for learning encoder representations."
>
> We wanted to keep the MBRL algorithms (DreamerV3 and TD-MPC2) as original as possible. In DreamerV3, the reconstruction term is an integral part of the world model's loss and can neither be replaced nor omitted easily. As in the original algorithm, we utilize reconstructions so that the world model is able to include information about the state also in future predictions. More specifically, reconstruction is not only needed to learn the encoder and decoder networks, but also to learn long-term dependencies via the dynamics prediction. Besides, [3] showed that Dreamer without reconstruction results in poorer overall performance. On the other hand, similar reasons motivated us to include TD-MPC2 as an additional algorithm; it does not involve reconstructions and is more decision-aware (i.e. relying more on rewards).
>
> > “Did the authors experiment with reconstructing the image observations from frozen PVRs in order to study the possibility of recovering image information from representations ?”
>
> We think the latter suggested experiment is an interesting approach for PVRs in combination with world models. Such experiments would take a similar line as our dynamics prediction accuracy experiments. Therefore, we did not perform such experiments and would leave them for future research. Besides, recovering image information is not necessarily an indicator for good reward or task-specific dynamics prediction performance since information needed for high rewards might not be present in the representation even if the overall reconstruction quality is good.
>
> > “Did the authors experiment with using representations of other layers than the final layer? “
>
> We conducted additional experiments during the rebuttal time by removing the transformer blocks of VC-1 partly (more specifically we ablated ⅔ and ⅓ of the 24 transformer blocks). Results can be seen in Figure 3 of the PDF to the Author Rebuttal above. Using ⅔ of VC-1 results in similar performance to the full transformer. This suggests that transformer blocks near the final one offer as much information as the final output. With only ⅓ of VC-1 the performance drops significantly. It seems that earlier representations do not offer enough information for the MBRL agent to perform better or similarly.
>
> > “For the in-distribution autoencoder PVR, what policy is used to generate the data used for pre-training ? Is it a random policy, a collection of policies to ensure diverse behaviors, or a pre-trained policy ?”
>
> We utilized data collected with a DreamerV3 policy during its training phase, including data from the exploration phases. We used an agent utilizing VC1 as an encoder. The motivation of using VC-1 as encoder was to enlarge the data distribution explored by the algorithm in comparison to a better performing agent training from scratch. This approach ensures that the autoencoder encounters a diverse data distribution similar to that of the other agents during their training, thereby ensuring also a fair comparison.
>
> __Weaknesses and Limitations:__
>
> > "... the authors only study the use of PVRs for continuous control tasks. It would have been interesting to study the application of PVRs to another domain based on discrete actions (Atari games, DeepMind Lab 3D environments, Minecraft...) or more aligned with data seen during PVR pre-training."
>
> We were able to generate additional results with a Miniworld [4] environment which supports discrete actions (see Figure 2 in the PDF attached to Author Rebuttal above). The new evidence supports our original claim that PVRs do not generally improve MBRL training and performance. Furthermore, we contend that the ManiSkill2 experiments are well aligned with the pre-training data of many PVRs (e.g., VC-1, R3M, VIP as promoted in the associated papers), and we were also interested in exploring their out-of-distribution performance in out-of-distribution domains like DMC.
>
>
> [1] Kumar, A., Raghunathan, A., Jones, R. M., Ma, T., & Liang, P. (2022). Fine-Tuning can Distort Pretrained Features and Underperform Out-of-Distribution. In International Conference on Learning Representations.
>
> [2] Majumdar, A., Yadav, K., Arnaud, S., Ma, Y. J., Chen, C., Silwal, S., … Meier, F. (2023). Where are we in the search for an Artificial Visual Cortex for Embodied Intelligence? In Thirty-seventh Conference on Neural Information Processing Systems.
>
> [3] Hafner, D., Lillicrap, T., Ba, J., & Norouzi, M. (2020). Dream to Control: Learning Behaviors by Latent Imagination. In International Conference on Learning Representations.
>
> [4] Chevalier-Boisvert, M., Dai, B., Towers, M., Perez-Vicente, R. D. L., Willems, L., Lahlou, S., … Terry, J. K. (2023). Minigrid & Miniworld: Modular & Customizable Reinforcement Learning Environments for Goal-Oriented Tasks. In Thirty-seventh Conference on Neural Information Processing Systems Datasets and Benchmarks Track.

---

> > ### Comment · Reviewer_32eA · 2024-08-10
> >
> > Thank you for taking the time to respond in the rebuttal,
> >
> > After reading the paper and rebuttal, I have decided to maintain my rating. The paper does show the inability of DreamerV3 and TD-MPC to achieve comparable or superior performance when using input observations from frozen PVRs. However, I think that it would benefit a lot from additional experiments on all tasks to better understand the possible causes of these findings: analysis using different layer features, fine-tuning of PVRs weights using reward and value loss gradients, analysis of accumulated reward/dynamics error on all tasks (not just Pendulum Swingup), and possible visualization of reconstructed video trajectories.
> >
> > The paper does provide interesting findings and could be used as reference for future works, but I think the analysis of the potential of PVRs is still insufficient for higher rating.

---

> > > ### Author Response · Authors · 2024-08-13
> > > **Thank you for your feedback to our rebuttal**
> > >
> > > Thank you for taking the time to discuss our rebuttal and provide your feedback.
> > >
> > > Our primary aim with this work was to benchmark the capabilities of the unchanged PVRs in MBRL and to highlight the limitations of PVRs. To this end, the type and the number of experiments conducted in our study is consistent with previous papers evaluating PVRs for policy learning (including new experiments from the rebuttal which also cover your suggested analysis of using different layer features). In addition to the reasons outlined in our rebuttal as to why more of the suggested experiments are out of scope of this paper, we argue that adding even more experiments also risk compromising the depth and quality of all aspects in the paper, especially since our paper has already reached the maximum page limit. Furthermore, conducting such an extensive study as ours involves significant effort as well as computational resources.
> > >
> > > By especially highlighting the limitations in our findings, we hope to save other researchers time and provide valuable insights for those exploring zero-shot PVR capabilities in future work. As also highlighted by yourself we also strongly believe that our work will serve as a useful reference point for other researchers.

---

### Official Review · Reviewer_DQ5X · 2024-07-08

**Soundness:** 3
**Presentation:** 3
**Contribution:** 3
**Rating:** 6
**Confidence:** 4

**Summary:**

This paper contains an extensive study on the potential benefits of pre-trained visual representations (PVRs) for model-based reinforcement learning (MBRL). Given the success of PVRs for model-free RL (MFRL), there are reasons to believe that PVRs are equally beneficial for MBRL, something that has not yet been explored to the same degree. Surprisingly, tests on two different RL architectures in combination with at least eight different PVR variants indicate that PVR-based solutions rarely perform better than representations learned from scratch when applied for MBRL. In most cases, it seems preferable not to use PVRs for MBRL, which should come as a surprise to many, given recent advances in applications based on PVRs.

**Strengths:**

Given the recent development of PVRs and the increasing number of successful applications of PVRs, it is interesting to know what the opportunities and limiting factors are for PVRs when adopted for training agents using reinforcement learning. A study with this as a focus, such as the one presented in this paper, should interest many. Besides the most important conclusion drawn in the presented study, i.e. the fact that the benefits of PVRs for MBRL can be questioned, it contains several experiments to paint a more complete picture.

It is concluded that the sample efficiency of MBRL agents rarely improves when supported by PVRs compared to those with representations learned from scratch. PVRs do not seem to make agents more robust of out-of-distribution (OOD) conditions either. It is also shown that the dynamics and reward prediction errors of RL agents are similar regardless of whether representations are learned from scratch or with PVRs. Also interesting is a study on the benefits of different properties of PVRs for generalization, where the most important property was shown to be data diversity during training, which might not be that surprising. The fact that language conditioning and the choice of transformer architecture have such a little impact might be more unexpected though.

**Weaknesses:**

From the experiments, it can be concluded that PVRs hardly benefit MBRL, but the paper does not really try to answer why there is a difference between MBRL and MFRL in this regard. Could it be that the way DreamerV3 and TD-MPC2 are modified in this study, the PVR modules $g_*$ are trained to preserve semantic information, while in effect also suppressing spatial information? Earlier studies on PVRs for MFRL, cited in the paper, do not seem to do this but instead keep a path that allows a policy to be trained with some spatial information left intact.

The authors seem surprised that language conditioning is not good for generalization, since it should provide semantically relevant features. Such a claim completely misses the fact that for e.g. manipulation it is often more important for a system to know where an object is located, how it is oriented and how large it is, than what class the object belongs to or any other semantic information. Most of our visual representations are explicitly trained to be invariant to transformations that might be particularly important for the task given to the agent.

It is worth noting that the experiments were conducted in simulation, not on real robots. The paper does not draw conclusions that go beyond what can be observed in the experiments, but the differences between simulated and real-world conditions are not really regarded. There are other potential benefits of MBRL when applied to real robots. With MBRL any action executed on the system can be exploited for pre-training, while MFRL is typically focused on only one particular type of task. It might be that this would not really affect the conclusions as expressed in the paper, but the experimental conditions between simulations and real-world experiments can be very large, The same is true regarding conclusions drawn from the OOD experiments. In the paper, the nature of randomizations is the same for both the ID and OOD sets, even if the two sets don’t overlap. If you were to do the experiments in the real world, the conditions would be very different.

**Questions:**

* What is believed to be the reason why the benefits of PVRs for MBRL seem so limited, in particular, given earlier success for MFRL?
* To what extent do the tested PVRs preserve spatial information relevant for tasks such as manipulation?

**Limitations:**

The fact that more experiments are needed to draw final conclusions is highlighted as a limitation, which is true indeed given the diversity of domains for which MBRL can be used and the computational demands for experiments in each such domain.

---

> ### Author Rebuttal · Authors · 2024-08-06
>
> __Weaknesses:__
>
> > "From the experiments, it can be concluded that PVRs hardly benefit MBRL, but the paper does not really try to answer why there is a difference between MBRL and MFRL in this regard."
>
> We believe that multiple objective mismatches in the different training phases, which are not all present in MFRL, result in a difference. We already discussed this in the first bullet point of the Author Rebuttal above and will include this in the revised paper.
>
> > “Could it be that the way DreamerV3 and TD-MPC2 are modified in this study, the PVR modules 𝑔∗ are trained to preserve semantic information, while in effect also suppressing spatial information?”
>
> Previous studies (described in the related work) sometimes use the PVRs in combination with additional task information (like proprioceptive or spatial information) or use the output of the PVRs only. Since we wanted to test the capabilities of the PVRs only, we refrain from using additional task information as input for the MBRL agents since it is hard to measure to what extent the PVRs or the additional information are used to solve the task. We are unsure why the reviewer believes that we actively suppress spatial information since we adapt DreamerV3 and TD-MPC2 only slightly? Many of the analyzed PVRs learn spatial information (e.g. VC-1, R2D2, Taskonomy Autoencoder).
>
> > “The authors seem surprised that language conditioning is not good for generalization, since it should provide semantically relevant features.”
>
> We would like to thank the reviewer for drawing our attention to this misunderstanding. We never wanted to convey this opinion but wanted to bring attention to the fact that many other approaches build up on language without utilizing additional spatial information (e.g. R3M and those mentioned in our paper). These papers utilize language in their approaches, asserting that it significantly enhances performance. We wanted to emphasize this perspective. We will revise this part of our paper for the final version.
>
> > “It is worth noting that the experiments were conducted in simulation, not on real robots. The paper does not draw conclusions that go beyond what can be observed in the experiments, but the differences between simulated and real-world conditions are not really regarded.”
>
> We appreciate the suggestion to train on real robots and will include this as an additional limitation in the paper. Conducting experiments on real robots is computationally demanding and time-consuming, particularly since our approach involves reinforcement learning rather than imitation learning. On the other hand, our approach was to benchmark the touted zero-shot capabilities of pre-trained visual models, rather than to devise a high-performant (MB)RL agent which performs well in real world tasks.
>
> __Questions:__
>
> > “What is believed to be the reason why the benefits of PVRs for MBRL seem so limited, in particular, given earlier success for MFRL?”
>
> We need to train a model of the environment as well as a reward predictor with those representations. The information preserved by the representations might not be enough to do both. (See also the related answer from the first weakness and Author Rebuttal above.)
>
> > “To what extent do the tested PVRs preserve spatial information relevant for tasks such as manipulation?”
>
> This individually depends upon the training loss of the PVR. While R3M and all versions of CLIP use semantic information via language, the losses of the other PVRs allow learning spatial information implicitly.
>
> __Limitations:__
>
> > "The fact that more experiments are needed to draw final conclusions is highlighted as a limitation, which is true indeed given the diversity of domains for which MBRL can be used and the computational demands for experiments in each such domain."
>
> We added an additional task from another environment domain (i.e. PickupObjects-v0 from Miniworld [1]). You can see the results in Figure 2 in the PDF attached to the Author Rebuttal at the top. The new evidence supports our original claim that PVRs do not generally improve MBRL training and performance.
>
> [1] Chevalier-Boisvert, M., Dai, B., Towers, M., Perez-Vicente, R. D. L., Willems, L., Lahlou, S., … Terry, J. K. (2023). Minigrid & Miniworld: Modular & Customizable Reinforcement Learning Environments for Goal-Oriented Tasks. In Thirty-seventh Conference on Neural Information Processing Systems Datasets and Benchmarks Track.

---

> ### Comment · Reviewer_DQ5X · 2024-08-13
>
> This reviewer wants to thank the authors for their informative comments on the points raised in the review. It is not suggested that the authors actively try to suppress spatial information. The question is rather why there is a difference between MFRL and MRBL with respect to the benefits of PVR. This reviewer suggests that if $g^*$ captures more semantic than spatial information, which might not be the case, there is no way for lost spatial information to be recovered, using the architectures in Figure 1. There might or might not be individual variations in implementations in terms of architectures and PVRs that better explain observed differences than any fundamental difference between MFRL and MRBL.

---

> > ### Author Response · Authors · 2024-08-13
> > **Thank you for your valuable comment**
> >
> > We thank the reviewer for the valuable comment and the helpful participation in the discussion.
> >
> > We agree that it is hard or even impossible to recover spatial information from semantic information (and vice versa) inside the world model/agent using the embedding of the PVRs only. Since it would be an individual paper to measure what kind of information is captured by the PVRs exactly, it is out of scope for our paper to analyze this more deeply.
> > On the other hand, we believe that the difference in PVR performance between MFRL and MBRL is not explained solely by the spatial or semantic information captured (or not captured) by the PVRs, as the type of information needed (spatial or semantic or both) ultimately depends on the task. Therefore, missing coverage of semantic and spatial information in the PVRs should affect MFRL and MBRL algorithms equally.
> >
> > Regarding the individual variations: We categorized the PVRs and their scores according to different characteristics of the PVRs in Section 4.3 of our paper to measure the effect of different implementation details and variations. We have already included a large and diverse set of PVRs in our study, incorporating different architectures, loss types, data modalities, and more. If the reason for our findings was related to one of these implementation details, we would likely have observed it in our results. Furthermore, the architectures used in MFRL and MBRL are quite similar, as most methods utilize CNNs, MLPs, and LSTMs/GRUs, particularly in the benchmarks we reference. DreamerV3 and TD-MPC2 also employ these architectures.
> >
> > We think (based on our results in the paper and the rebuttal) that the aforementioned performance difference stems from the observation that reward-related information is not sufficiently captured in the representations to learn accurate reward prediction models (as needed by MBRL algorithms in contrast to MFRL methods).

---

### Official Review · Reviewer_HgP2 · 2024-07-12

**Soundness:** 2
**Presentation:** 2
**Contribution:** 2
**Rating:** 5
**Confidence:** 3

**Summary:**

This paper conducts a thorough set of experiments to evaluate the performance of pretrained visual representations (PVR) in model-based reinforcement learning. Empirical results show that PVR performs worse than learning from scratch, which is possibly due to the large reward prediction error.

**Strengths:**

- This paper shows how pre-trained visual representations perform in MBRL, which has been relatively unexplored in previous research.
- A thorough set of experiments reveals that pre-trained representations perform worse than learning from scratch, which is surprising and could inspire future research to analyze this phenomenon further.

**Weaknesses:**

- This paper gives some possible reasons for the performance degradation of pre-trained representations. However, it lacks sufficient evidence to further support the claims. For example, Why does the reward prediction accuracy outweigh the dynamic prediction accuracy? Why do pre-trained representations lead to information loss about reward prediction? I encourage the author to dig deeper into the specific properties of pre-trained representations used in MBRL.

**Questions:**

- I think the performance degradation of PVR is mainly due to the fact that they are not trained on the experimental data used in this paper, as Autoencoder performs similarly compared to learning from scratch. Therefore, could you fine-tune the pre-trained visual encoder and then evaluate the performance? I guess slight fine-tuning can boost the performance a lot.
- Why does VC-1 perform almost the best in maniskill (see figure 3 and figure 4)? Is VC-1 specifically pre-trained on these manipulation data?
- Another possible reason is that models pre-trained on large datasets are usually hard to transfer to low-data regimes. See UniSim [1] for details.

[1] Yang et al., Learning Interactive Real-World Simulators, ICLR 2024

**Limitations:**

Limitations have been discussed in this paper.

---

> ### Author Rebuttal · Authors · 2024-08-06
>
> __Weaknesses:__
>
> > “Why does the reward prediction accuracy outweigh the dynamic prediction accuracy?”
>
> Our Figures 6 and 7 show that there exist differences in the dynamics as well as reward prediction accuracies between the trained models. The calculated correlations indicate that more subtle differences in reward predictions compared to dynamics predictions lead to poorer performing policies. Even though all models learn to predict the dynamics and are incentivized to do so via the losses of DreamerV3 and TD-MPC2, the best measure of task performance is the reward. Thus, its prediction quality is intuitively stronger correlated to task performance (as shown in our paper with the mentioned correlations). Since the predictions are directly used to update the policy, predictions must be accurate.
>
> > “Why do pre-trained representations lead to information loss about reward prediction?”
>
> We argue that those representations lose information about rewards during pre-training since they are most often trained with different training objectives than later used during the MBRL phase. That such an objective mismatch influences the training was already shown for MBRL itself, as MBRL methods like Dreamer and TD-MPC need to learn the dynamics and a policy with conflicting objectives. The additional mismatch in objectives between PVRs and MBRL is now an additional factor. In general, the PVRs are trained to compress the given information content (i.e. information bottleneck theory) and are not incentivized to capture information in the images relevant for a possible reward function.
>
> __Questions:__
>
> > “Therefore, could you fine-tune the pre-trained visual encoder and then evaluate the performance?”
>
> We actively decided to omit fine-tuning experiments since the original promise of many of the used representations is to perform (near) SOTA on downstream tasks without additional fine-tuning and we wanted to examine these claims. Fine-tuning is therefore not the focus of our paper. Furthermore, correctly fine-tuning might not be an easy task since it can destroy parts of the representation needed for good generalization performance [1]. Ultimately, we believe that (if done right) fine-tuning can boost performance for each of the representations. But we are unsure if the performance boost would outperform training from scratch. This aligns with mixed results from end-to-end fine-tuning in the original VC-1 paper [2]. Furthermore, fine-tuning before the RL phase would also result in an unfair advantage compared to training the representation from scratch. But we believe that the question of how to adequately fine-tune such representations for MBRL and RL in general is nevertheless important and should be the topic of future research.
>
> > “Why does VC-1 perform almost the best in maniskill (see figure 3 and figure 4)? Is VC-1 specifically pre-trained on these manipulation data?”
>
> VC-1 was trained on a combination of datasets with a connection to manipulation and navigation. More specifically, VC-1 uses Ego4D and similar human manipulation as well as navigation data (but never robot manipulation data). We hypothesize that in this case the domain gap is smaller for ManiSkill2 due to the connection of the pre-training data to manipulation. Therefore, a combination of manipulation related data seems to be somewhat beneficial; even if it's observations of humans.
>
> > “Another possible reason is that models pre-trained on large datasets are usually hard to transfer to low-data regimes. See UniSim [1] for details.”
>
> We thank you for pointing out this important part of UniSim, since it is relevant for our conclusion. The paper states "During joint training of the UniSim on diverse data, we found that naïvely combining datasets of highly varying size can result in low generation quality in low-data domains.". As already mentioned, VC-1 uses a curated combination of datasets which supports this claim of UniSim.
>
> [1] Kumar, A., Raghunathan, A., Jones, R. M., Ma, T., & Liang, P. (2022). Fine-Tuning can Distort Pretrained Features and Underperform Out-of-Distribution. In International Conference on Learning Representations.
>
> [2] Majumdar, A., Yadav, K., Arnaud, S., Ma, Y. J., Chen, C., Silwal, S., … Meier, F. (2023). Where are we in the search for an Artificial Visual Cortex for Embodied Intelligence? In Thirty-seventh Conference on Neural Information Processing Systems.

---

> > ### Comment · Reviewer_HgP2 · 2024-08-08
> > **Thanks for your response**
> >
> > After reading your rebuttal, I've decided to maintain my rating. It is still hard to determine the accurate answer to why PVR is useless for MBRL. I believe that future research should focus on how to properly incorporate PVR for better performance, like in many other machine learning scenarios. However, this paper is still limited to providing some possible approaches to addressing this challenge. I encourage the authors to study this problem further and dig deeper. That will be a high-quality paper.

---

> > > ### Author Response · Authors · 2024-08-12
> > > **Thank you for your feedback to our rebuttal**
> > >
> > > Thank you for your feedback. We appreciate your encouragement and look forward to building on this work in future studies.
> > >
> > > Our primary goal with this paper is to draw attention to the issue and highlight the limitations of PVRs in MBRL. We agree that finding effective ways to incorporate PVRs for better performance is crucial, and we plan to address this in our future research. Attempting to tackle both the identification of the problem and its solution in a single paper would have risked compromising the depth and quality of each aspect. We think this paper alone is valuable to the community as it serves as a useful reference point for future research on the development of PVRs for generalist embodied agents. Furthermore, the presentation of such a paper at NeurIPS is important as it encourages critical thinking and discussion about our current approaches in developing PVRs.

---

### Official Review · Reviewer_mY2R · 2024-07-13

**Soundness:** 3
**Presentation:** 3
**Contribution:** 3
**Rating:** 6
**Confidence:** 4

**Summary:**

Pre-trained Visual Representations (PVRs) has been widely applied to many domains to improve OOD generalization and sample efficiency, including model-free reinforcement learning (RL). This paper explores the application of PVRs in model-based RL, which has not been done. Experiments on two suite of simulated control environments (Deepmind control and Maniskill2) show that current PVRs are outperformed by representations learned from scratch in terms of sample efficiency and OOD generalization. The paper further analyzes the reward error and dynamics prediction error of various PVRs and provides meaningful insights.

**Strengths:**

- The paper is well written and easy to follow. The research questions that the paper investigate are clearly stated. The proposed benchmark is well-motivated.
- The proposed benchmark has great potential in advancing research in developing better PVRs for MBRL.
- The paper provides sufficient detail for the experiments allowing easier reproduction of the results.
- The experiments includes a large suite of popular PVRs, improving the credibility of the conclusions.

**Weaknesses:**

- The number of algorithms in each property category is small, which means that the performance of one algorithm can easily affect the average performance of the category, making the comparison of different categories (Figure 5) less significant. For example, the "sequential data" category and the "temporal loss" category only differ in one method (VC-1), and attributing the difference caused by this one method to the sequential data property might not be correct, since there are many other differences between VC-1 and R3M/VIP.

**Questions:**

- Different tasks can have very different reward functions even for the same environment, and we have no clue on what the reward function is when pre-training self-supervised visual representations. Then, why should we expect a single PVR to perform well on learning good reward models for all kinds of tasks? In other words, is the task of building PVRs that can learn good reward models a somewhat intractable one?

**Limitations:**

The proposed benchmark only evaluated on two suite of simulated control environments, which, as the authors also pointed out in the paper, is (1) somewhat insufficient, and (2) very challenging for the PVRs concerned because many of them are never trained on similar data distributions (especially for DMC). Therefore, saying that PVRs are "ineffective" or MBRL seems like an overstatement to me. Strong claims need strong evidence and the evidence provided in the paper is in my opinion insufficient. Of course I still appreciate the value of the proposed benchmark in that it brings the problem of applying PVRs to MBRL to the community's attention, but I do not agree with putting such an assertive statement as the title of the paper.

---

> ### Author Rebuttal · Authors · 2024-08-06
>
> __Weaknesses:__
>
> > “The number of algorithms in each property category is small, which means that the performance of one algorithm can easily affect the average performance of the category, making the comparison of different categories (Figure 5) less significant. For example, the "sequential data" category and the "temporal loss" category only differ in one method (VC-1), and attributing the difference caused by this one method to the sequential data property might not be correct, since there are many other differences between VC-1 and R3M/VIP.”
>
> We integrated the mentioned categories “Sequential Data” and “Temporal Loss” since VC-1 is trained on sequential data but does not make use of this characteristic. While we agree that your statement is true for these two categories, the PVRs of the other categories represent a more diverse set of representations. In the other categories, especially “ViT” and “Diverse Data”, the performance differences are significant. Therefore, we will revise our paper accordingly and will remove the category “Temporal Loss” from our findings since the overlap between this category and “Sequential Data” is too large. Thank you for this tip.
>
> __Questions:__
>
> > “Then, why should we expect a single PVR to perform well on learning good reward models for all kinds of tasks?”
>
> This expectation was established by various other papers referenced in our relevant work and by representations like VC-1, VIP, etc. We do not want to advocate the use of PVRs in MBRL per se. Our work is a benchmark of PVRs following the zero-shot idea established by previous works on MFRL and imitation learning. These works argue that PVRs can be helpful in downstream reinforcement learning tasks because the representations saw vast amounts of different data during training and/or were trained with losses capturing some form of value. The reason why we created the presented benchmark is precisely to challenge these claims. We do not make any claims regarding the capability of the investigated PVRs to capture features relevant for arbitrary reward models for all kinds of tasks. Our results highlight that there is not a single PVR that outperforms from scratch training in every environment.
>
> > “In other words, is the task of building PVRs that can learn good reward models a somewhat intractable one?”
>
> It is most likely intractable to train a single PVR capturing reward information for all possible tasks. But our results suggest that e.g. diverse but curated datasets might help for specific tasks. VC-1 especially shows that a combination of diverse datasets taking e.g. manipulation aspects into account is an important element for a PVR that is used for manipulation tasks. However, VC-1 still cannot solve all tasks as effectively as learning a representation from scratch with MBRL (i.e. solving DMC tasks).
>
> Furthermore, both learning phases (pre-training of representations and downstream RL policy training using such PVRs) are heavily influenced by an objective mismatch since the pre-training objective and the RL training objective usually differ much. PVRs are not incentivized to capture reward related information from the images and might discard that information during learning whereas RL training objectives are driven by a reward function (see UMAPs in Figure 1 in PDF of Author Rebuttal).
>
> __Limitations:__
>
> > “...because many of them are never trained on similar data distributions”
>
> Indeed, the PVRs are often not trained on data distributions similar to our experiments. But we want to mention that many of the original papers of the PVRs used actually claim that their representations work on DMC and robotics environments like ManiSkill2 or even provide experiments supporting those claims (see VC-1, VIP, etc.). E.g. VIP was trained on Ego4D only. The related paper [1] shows that algorithms using VIP can solve robotic manipulation tasks. VC-1 was trained on diverse control data and was shown to work on DMC as well. The same paper includes experiments showcasing that VIP and R3M enable RL policies to be SOTA in DMC. The Taskonomy representations were used to solve tasks of the VizDoom environments and robotic manipulation tasks. We used our presented environments to show that those claims do not hold for MBRL.
>
> > “Of course I still appreciate the value of the proposed benchmark in that it brings the problem of applying PVRs to MBRL to the community's attention, but I do not agree with putting such an assertive statement as the title of the paper.”
>
> We understand your concerns about the assertiveness of our claims and title. With an additional navigation experiment from the Miniworld [2] environment we will improve the paper to temper these claims, ensuring they more accurately reflect the evidence presented. You can see those results in Figure 2 in the PDF of the Author Rebuttal above. The new evidence still supports our original hypothesis.
> Our intention in choosing a somewhat provocative title is to highlight the above-mentioned potential limitation of current PVR approaches when applied to MBRL (as you mention yourself) and to challenge the common assumption that PVRs *generally* improve RL training. We do not claim that PVRs are worse in every environment, but many PVRs are presented as general foundation models, implying they should perform well on DMC as well. However, they do not, which is reflected in our title.
>
> [1] Ma, Y. J., Sodhani, S., Jayaraman, D., Bastani, O., Kumar, V., & Zhang, A. (2023). VIP: Towards Universal Visual Reward and Representation via Value-Implicit Pre-Training. In The Eleventh International Conference on Learning Representations.
>
> [2] Chevalier-Boisvert, M., Dai, B., Towers, M., Perez-Vicente, R. D. L., Willems, L., Lahlou, S., … Terry, J. K. (2023). Minigrid & Miniworld: Modular & Customizable Reinforcement Learning Environments for Goal-Oriented Tasks. In Thirty-seventh Conference on Neural Information Processing Systems Datasets and Benchmarks Track.

---

> > ### Comment · Reviewer_mY2R · 2024-08-08
> >
> > I thank the authors for addressing my questions and concerns. I increased the confidence score to 4.
> >
> > Considering the overall contributions and limitations of the paper (including ones that the other reviewers pointed out), I will keep my other ratings.

---

> > > ### Author Response · Authors · 2024-08-13
> > > **Thank you for increasing your confidence score**
> > >
> > > Thank you for taking the time to review our paper and for increasing the confidence score.
> > >
> > > We value your feedback and are grateful for your insights. We will integrate your suggestions, as discussed in the rebuttal, in the final version of the paper.

---

### Author Rebuttal · Authors · 2024-08-06

We thank the Reviewers for their thorough feedback. We appreciate the detailed suggestions and the recognition of the value that our results bring to the community. We were keen to integrate the Reviewers valuable feedback into our paper and will make the revisions to the paper accordingly. Here is a summary of changes and answers:

- Reviewer HgP2 raised questions regarding the underlying causes of the poor performance in reward prediction. We ultimately believe that this is due to multiple objective mismatches which are present between the pre-training of the visual representations (PVRs) and the actual MBRL training phase, as well as in MBRL itself where we have a mismatch between dynamics learning and policy learning [1]. Especially, the first mentioned objective mismatch complicates the transfer of reward information, since PVRs are trained to compress information (information bottleneck theory) and as such the training might not capture reward-relevant data from the pre-training images. On the other hand, MBRL methods like DreamerV3 and TD-MPC2 heavily rely on reward information in their objective. We added UMAP embeddings (see Figure 1 of the attached PDF) showing that reward information is more closely embedded in representations which are learned by the agents completely from scratch. Regarding previous results we additionally want to mention that reward information was often irrelevant in previous benchmarks based on imitation learning.
- We got several reviewer questions regarding fine-tuning some of the PVRs on task-specific data (HgP2;32eA). Since we wanted to test the touted zero-shot capabilities of the PVRs, we consciously decided against additional fine-tuning experiments. Fine-tuning is therefore not a focus of our paper. Furthermore, since fine-tuning causes its own difficulties and challenges (e.g. it can distort features [2]), we want to leave those research questions for future research.
- Reviewer 32eA suggests ablating the PVRs as it was already done in previous benchmarks. Therefore, we conducted an experiment with DreamerV3 and VC-1 removing ⅓ and ⅔ of the 24 transformer blocks of VC-1. Results can be seen in Figure 3 of the attached PDF. Outputs of earlier transformer blocks do not exceed the performance of the full representation.
- One way to gain further insights is to expand task domains beyond DMC and ManiSkill2 as also pointed out by reviewers mY2R and DQ5X. We already acknowledged this in our limitations. Both mentioned domains support continuous actions only. We now included an additional navigation experiment from Miniworld [3]. Due to the discrete action-space and the short rebuttal time, we were only able to perform the experiment with DreamerV3 and a selected number of PVRs. TD-MPC2 does not support discrete action spaces. The results can be seen in Figure 2 of the attached PDF. The new evidence supports our original claim that PVRs do not generally improve MBRL training and performance. Similar to the other experiments, DreamerV3 agents trained from scratch are more sample efficient and performant compared to agents using PVRs. Only VC-1 is able to perform comparably.

On the other hand, Reviewers highlighted the clarity and thoroughness of our paper (mY2R;32eA), the novel exploration of PVRs in MBRL (HgP2), and the well-motivated benchmark (including out-of-distribution evaluation) that advances research in this area (mY2R;32eA). Reviewer DQ5X highlighted that such a “study with this as a focus, such as the one presented in this paper, should interest many.”. The reviewers also valued our surprising findings that pre-trained representations often perform worse than learning from scratch (HgP2;32eA;DQ5X) and the large number of PVRs analyzed (mY2R;32eA).

We sincerely hope that the changes and answers have addressed reviewer concerns. If so, we kindly request that you consider revising your scores. Please let us know if you have any further comments!

[1] Lambert, N., Amos, B., Yadan, O., & Calandra, R. (2020). Objective Mismatch in Model-based Reinforcement Learning. In Learning for Dynamics and Control (pp. 761–770). PMLR.

[2] Kumar, A., Raghunathan, A., Jones, R. M., Ma, T., & Liang, P. (2022). Fine-Tuning can Distort Pretrained Features and Underperform Out-of-Distribution. In International Conference on Learning Representations.

[3] Chevalier-Boisvert, M., Dai, B., Towers, M., Perez-Vicente, R. D. L., Willems, L., Lahlou, S., … Terry, J. K. (2023). Minigrid & Miniworld: Modular & Customizable Reinforcement Learning Environments for Goal-Oriented Tasks. In Thirty-seventh Conference on Neural Information Processing Systems Datasets and Benchmarks Track.

---

### Decision · Program_Chairs · 2024-09-25

**Decision:**

Accept (poster)

**Comment:**

The paper's central hypothesis is to provide experimental evidence that PVR is ineffective for MBRL and learning representations from scratch is as good (or even better) than using pre trained features. The reviewers generally agree that the presented experiments are barely enough to demonstrate this phenomenon, while suggesting a few improvements. Firstly, conducting more experiments is required to assess the generality of this hypothesis -- including but not limited to increasing the number of methods in each property category. Secondly, further analysis is needed to understand the underlying reasons for this observation. I agree with the reviewers that these are valid weaknesses of the paper. Particularly, lack of enough analysis on "why" PVR is not beneficial to MBRL can be explored further and with additional experiments and analysis, it may become apparent that the findings are limited to the specific environments and methods tested. However, I believe the paper's primary conclusions are significant and thought-provoking enough to warrant consideration as a valuable contribution to MBRL. Therefore, I recommend accepting the paper as a poster.